# Plasma pentosidine levels are associated with prevalent fractures in patients with chronic liver disease

Chisato Saeki[1,2]*, Mitsuru Saito[3], Tomoya Kanai[1,2], Masanori Nakano[1,2], Tsunekazu Oikawa[1], Yuichi Torisu[1,2], Masayuki Saruta[1], Akihito Tsubota[4]*

**1** Division of Gastroenterology and Hepatology, Department of Internal Medicine, The Jikei University School of Medicine, Tokyo, Japan, **2** Division of Gastroenterology, Department of Internal Medicine, Fuji City General Hospital, Fuji city, Shizuoka, Japan, **3** Department of Orthopaedic Surgery, The Jikei University School of Medicine, Tokyo, Japan, **4** Core Research Facilities, Research Center for Medical Science, The Jikei University School of Medicine, Tokyo, Japan

* chisato@jikei.ac.jp (CS); atsubo@jikei.ac.jp (AT)

**Data Availability Statement:** All relevant data are within the manuscript and its Supporting Information files.

## Abstract

### Aim

Osteoporotic fractures negatively impact health-related quality of life and prognosis. Advanced glycation end products (AGEs) impair bone quality and reduce bone strength. The aim of this study was to determine the relationship between plasma levels of pentosidine, a surrogate marker for AGEs, and prevalent fractures in patients with chronic liver disease (CLD).

### Methods

This cross-sectional study included 324 patients with CLD. Vertebral fractures were evaluated using lateral thoracolumbar spine radiographs. Information on prevalent fractures was obtained through a medical interview, medical records, and/or radiography. The patients were classified into low (L), intermediate (I), and high (H) pentosidine (Pen) groups based on baseline plasma pentosidine levels.

### Results

Of the 324 patients, 105 (32.4%) had prevalent fractures. The prevalence of liver cirrhosis (LC) and prevalent fractures significantly increased stepwise with elevated pentosidine levels. The H-Pen group had the highest prevalence of LC (88.6%, $p < 0.001$) and prevalent fractures (44.3%, $p = 0.007$), whereas the L-Pen group had the lowest prevalence of LC (32.1%, $p < 0.001$) and prevalent fractures (21.0%, $p = 0.007$). Multiple logistic regression analysis identified pentosidine as a significant independent factor related to prevalent fractures (odds ratio = 1.069, $p < 0.001$). Pentosidine levels increased stepwise and correlated with liver disease severity. They were markedly high in patients with decompensated LC. In multiple regression analysis, liver functional reserve factors (total bilirubin, albumin, and prothrombin time-international normalized ratio) significantly and independently correlated with pentosidine levels.

**Funding:** The authors received no specific funding for this work.

**Competing interests:** The authors have declared that no competing interests exist.

## Conclusions

Plasma pentosidine was significantly associated with prevalent fractures and liver functional reserve in patients with CLD. Pentosidine may be useful in predicting fracture risk and should be closely followed in CLD patients with advanced disease.

## Introduction

Bone health is a global concern, because bone disorders are related to poor health-related quality of life, disability, and prognosis [1]. Osteoporosis, a metabolic bone disease, and consequent fragility fractures are common extrahepatic complications in patients with chronic liver disease (CLD) [2–9]. A meta-analysis of six case–control studies demonstrated that the prevalence of osteoporosis in patients with liver cirrhosis (LC) was significantly higher than that in healthy individuals [34.7% vs. 12.8%; odds ratio (OR) = 2.52] [5]. Another meta-analysis of seven studies revealed a significant association between CLD and the risk of osteoporotic fractures, with a pooled OR of 2.13 [8]. Specifically, vertebral fractures are more common in patients with CLD. These vertebral fractures are often undetected [4, 9] but cause impaired physical function, immobility, and sarcopenia [10, 11]. Therefore, appropriate assessment of bone metabolism status and osteoporotic fracture risk is essential for patients with CLD.

Both bone mineral density (BMD) and bone quality are crucial for bone strength [12, 13]. Bone quality is characterized by structural properties (microarchitecture) regulated by bone remodeling and material properties (collagen cross-link formation) of the bone [12, 13]. Non-enzymatic cross-links are typified by advanced glycation end products (AGEs) induced by non-enzymatic glycation, oxidation, or glycoxidation. AGEs impair the function of osteoblasts and cause bone fragility [12, 13]. Pentosidine is a surrogate marker for AGEs that accumulate in bone due to advanced age or diseases, such as chronic kidney disease (CKD) and diabetes [12–16]. Indeed, high levels of serum and urinary pentosidine are independent factors for osteoporotic fractures in patients with diabetes and postmenopausal women [17–20]. Thus, pentosidine is a useful marker for appraising fracture risk in these diseases/conditions. However, the relationship between pentosidine levels and fractures in patients with CLD is unclear. In the present study, we aimed to determine the association between plasma pentosidine levels and prevalent fractures in patients with CLD.

## Materials and methods

### Study design and patients

This cross-sectional study included 324 consecutive patients with CLD who presented to Fuji City General Hospital (Shizuoka, Japan) between October 2017 and April 2020. The inclusion criteria were as follows: (1) CLD patients with some etiology; (2) availability of BMD measurements using dual-energy X-ray absorptiometry (DEXA); (3) availability of data on vertebral fractures evaluated with lateral spinal radiographs; and (4) availability of information on prevalent fractures obtained through the medical interview, medical records, and/or radiography. The exclusion criteria were as follows: (1) fractures due to pathological processes, such as bone metastasis and cancer; (2) prolonged administration (>3 months) of glucocorticoids (>5 mg) within 12 months before study entry; and (3) CLD with two or more etiologies. This study protocol was briefly explained in the leaflet or verbally to almost all outpatients and inpatients with CLD. Those who were supportive of this study and met the inclusion criteria were explained on the research contents in the briefing paper. After providing written informed consent, they were finally enrolled in the study. Our hospital is located in Fuji City, which has

a population of approximately 250,000, near Mt. Fuji, and is the only community-based core hospital (520-bed capacity) in and around Fuji City. Therefore, this study cohort might have heterogeneous clinical characteristics but could represent the actual situation in the community-based and real-world clinical settings. This study was approved by the ethics committee of Fuji City General Hospital (approval No. 162) and carried out in accordance with the Declaration of Helsinki. Written informed consent was obtained from all patients.

## Diagnosis and definition of disease conditions

Patients were diagnosed with CLD if they had persistently elevated liver enzymes for ≥6 months, reflecting liver necroinflammation, and/or histopathological findings compatible with CLD on liver biopsy specimens, irrespective of the etiology. If patients had CLD with current and/or past history of heavy alcohol consumption (>3 units/day) and without other etiologies, they were diagnosed with alcoholic liver disease (ALD) [21]. Current drinking and smoking were defined as continuous heavy alcohol consumption and smoking within at least 1 month before the survey. Meanwhile, patients were diagnosed with LC according to the laboratory tests, morphological assessment with imaging (ultrasonography, computed tomography, and/or magnetic resonance), and presentation of portal hypertension (such as esophageal/gastric varices and ascites). The severity of LC was evaluated according to the Child–Pugh scoring system, which consists of the following five clinical components: levels of total bilirubin, albumin, and prothrombin time (PT), grade of hepatic encephalopathy, and degree of ascites. Each component was scored from 1 to 3, and all five component scores were summed to obtain the total score, which was classified into class A (5–6 points), B (7–9 points), and C (10–15 points), with class C being the most severe [22]. CKD was defined as an estimated glomerular filtration rate (eGFR) of <60 mL/min/1.73 m$^2$ for at least 3 months [23]. Vitamin D deficiency was defined as serum 25-hydroxyvitamin D [25(OH)D] levels ≤20 ng/mL [24].

## Fracture assessment

Prevalent fragility fractures were defined as a history of fractures of the vertebra, proximal portion of the humerus and femur, distal portion of the radius, lower extremity, rib, or pelvis, which occurred after the age of 40 years [17]. Prevalent vertebral fractures, including asymptomatic fractures diagnosed only by radiography, were semi-quantitatively assessed using lateral thoracolumbar spine radiographs [25]. Patients who had fragility fractures at the time of study entry were also included.

## Diagnosis of osteoporosis

BMD was measured at the lumbar spine (L2–L4), femoral neck, and total hip using DEXA (PRODIGY, GE Healthcare, Madison, WI, USA). Osteoporosis (T-score ≤ −2.5) was diagnosed on the basis of the World Health Organization criteria [26].

## Laboratory assessment

Routine laboratory tests (e.g., serum total bilirubin, albumin, and creatinine) were measured using automated biochemical analyzer (TBA-2000FR; Toshiba, Tokyo, Japan). Prothrombin time-international normalized ratio (PT-INR) was measured using a thromboplastin reagent (Coagpia PT-N; Sekisui Medical, Tokyo, Japan). The eGFR was calculated using the following formula: eGFR (mL/min/1.73 m$^2$) = 194 × Creatinine −1.094 × Age $^{-0.287}$ (× 0.739 for women).

Serum 25(OH)D was measured using a chemiluminescent immunoassay (Liaison 25-hydroxyvitamin D Total; Hitachi Chemical Diagnostics Systems, Tokyo, Japan). Mac-2 binding

protein glycosylation isomer (M2BPGi) was measured as a hepatic fibrosis marker using a sandwich enzyme-linked immunosorbent assay (ELISA) with Wisteria floribunda lectin-recognizing carbohydrate chains (HISCL-2000i; Sysmex, Hyogo, Japan), which were standardized and converted to a cutoff index according to the manufacturer's specified formula. Insulin-like growth factor-1 (IGF-1) was measured using an immunoradiometric assay (IGF-1 IRMA; Fujirebio, Tokyo, Japan). Plasma pentosidine levels were measured using an ELISA (FSK pentosidine ELISA kit; Fushimi Pharmaceutical, Kagawa, Japan), as previously described [18, 27]. Briefly, we incubated 50 μL of plasma samples with pronase at 55˚C for 1.5 h and then heated them in boiling water for 15 min to inactivate the enzyme reaction. The pretreated plasma samples were incubated with pentosidine-specific rabbit antibody at 37˚C for 1 h, added with peroxidase-labeled goat anti-rabbit IgG polyclonal antibody, and then re-incubated at room temperature for 1 h. We stopped the reaction 10 min after adding a color-developing reagent and measured the absorbance at 450 nm (main wavelength) and 630 nm (reference wavelength).

## Classification based on the plasma pentosidine levels

The median pentosidine level for all patients was 0.0598 (interquartile range, 0.0465–0.0886) μg/mL. The patients were divided into three groups according to the first and third quartiles (S1 Fig) for pentosidine levels, as follows: (1) the low pentosidine (L-Pen) group had pentosidine levels ≤0.0465 μg/mL (first quartile); (2) the intermediate pentosidine (I-Pen) group had pentosidine levels between 0.0465 μg/mL and 0.0886 μg/mL (third quartile); and (3) the high pentosidine (H-Pen) group had pentosidine levels ≥0.0886 μg/mL.

## Statistical analysis

Continuous and categorical variables are presented as medians (interquartile ranges) and relative frequencies (percentages), respectively. The Mann–Whitney U test and Kruskal–Wallis test followed by the Steel–Dwass post-hoc test were used to estimate differences in continuous variables between two groups and among three or more groups, respectively. The chi-squared test was used to estimate the differences in categorical variables between groups. The Cochran–Armitage trend test was used to assess whether a trend was present between a variable with two categories and a variable with multiple categories. Univariate and multiple logistic regression analyses were performed to identify significant and independent factors related to prevalent fractures. To predict the presence or absence of prevalent fractures, the receiver operating characteristic (ROC) curve of pentosidine was drawn and the optimal cutoff value was determined by the Youden index [28]. The Spearman's rank correlation test was performed to investigate correlations between plasma pentosidine and continuous variables. Factors that were significantly and independently related to plasma pentosidine levels were determined by multiple regression analysis. Statistical analyses were performed using SPSS version 26 (IBM Japan, Tokyo, Japan), with $p < 0.05$ indicating statistical significance.

# Results

## Patient baseline characteristics

The baseline clinical characteristics of the 324 patients with CLD are presented in Table 1. There were 159 men (49.1%) and 165 women (50.9%), with a median age of 69.0 (59.0–76.0) years. Among the 165 women, 155 (93.9%) were postmenopausal with no hormone supplementation. The numbers of patients diagnosed with LC, diabetes, and CKD were 188 (58.0%), 85 (26.2%), and 137 (42.3%), respectively. The etiologies were as follows: hepatitis B virus

**Table 1. Comparison of clinical characteristics between patients with and without prevalent fractures.**

| Variable | All patients | Fracture | Non-fracture | *p* value |
|---|---|---|---|---|
| Patients, n (%) | 324 | 105 (32.4) | 219 (67.6) | |
| Man, n (%) | 159 (49.1) | 49 (46.7) | 110 (50.2) | 0.548 |
| Age (years) | 69.0 (59.0–76.0) | 75.0 (70.0–80.0) | 66.0 (56.0–73.0) | < 0.001 |
| BMI (kg/m²) | 23.1 (20.8–26.0) | 22.2 (20.1–25.4) | 23.6 (21.0–26.1) | 0.012 |
| Current smoking, n (%) | 86 (26.5) | 22 (21.0) | 64 (29.2) | 0.115 |
| Current drinking, n (%) | 36 (11.1) | 9 (8.6) | 27 (12.3) | 0.314 |
| Menopause, n (%) | 155 (93.9) | 56 (100) | 99 (90.8) | 0.019 |
| Diabetes mellitus, n (%) | 85 (26.2) | 29 (27.6) | 56 (25.6) | 0.695 |
| Chronic kidney disease, n (%) | 137 (42.3) | 53 (50.5) | 84 (38.4) | 0.039 |
| Liver cirrhosis, n (%) | 188 (58.0) | 72 (68.6) | 116 (53.0) | 0.008 |
| Child-Pugh B+C, n (%) | 77 (41.0) | 29 (40.3) | 48 (41.4) | 0.881 |
| Etiology | | | | |
| HBV/HCV/AL/PBC/other, n | 46/99/63/62/54 | 10/35/21/18/21 | 36/64/42/44/33 | 0.384 |
| Total bilirubin (mg/dL) | 0.7 (0.5–1.0) | 0.7 (0.5–1.0) | 0.8 (0.6–1.1) | 0.099 |
| Albumin (g/dL) | 3.9 (3.5–4.3) | 3.9 (3.5–4.2) | 4.0 (3.5–4.3) | 0.164 |
| Prothrombin time INR | 1.05 (0.98–1.16) | 1.06 (0.99–1.17) | 1.05 (0.97–1.15) | 0.434 |
| Creatinine (mg/dL) | 0.8 (0.7–1.0) | 0.8 (0.7–1.1) | 0.8 (0.7–1.0) | 0.289 |
| eGFR (mL/min/1.73m²) | 64 (51–76) | 59 (47–73) | 65 (54–78) | 0.012 |
| M2BPGi (C.O.I) | 1.67 (0.89–4.35) | 2.10 (1.36–4.64) | 1.56 (0.72–4.35) | 0.003 |
| IGF-1 (ng/mL) | 63 (45–86) | 56 (42–73) | 67 (47–95) | 0.001 |
| 25(OH)D (ng/mL) | 13.4 (9.7–17.7) | 13.3 (10.4–17.8) | 13.4 (9.7–17.7) | 0.975 |
| Vitamin D deficiency, n (%) | 280 (87.0) | 94 (89.5) | 186 (85.7) | 0.341 |
| Pentosidine (μg/mL) | 0.0598 (0.0465–0.0878) | 0.0678 (0.0506–0.1029) | 0.0582 (0.0443–0.0820) | 0.004 |
| Lumbar spine BMD (g/cm²) | 1.07 (0.90–1.21) | 0.98 (0.84–1.13) | 1.11 (0.94–1.24) | < 0.001 |
| Femoral neck BMD (g/cm²) | 0.76 (0.66–0.88) | 0.69 (0.61–0.78) | 0.81 (0.70–0.90) | < 0.001 |
| Total hip BMD (g/cm²) | 0.83 (0.71–0.94) | 0.72 (0.63–0.83) | 0.86 (0.76–0.97) | < 0.001 |
| Osteoporosis, n (%) | 103 (31.8) | 59 (56.2) | 44 (20.1) | < 0.001 |

Values are presented as medians (interquartile ranges) or relative frequencies (percentages). Statistical analysis was performed using the chi-squared test or the Mann-Whitney U test, as appropriate. 25(OH)D, 25-hydroxyvitamin D; AL, alcohol; BMD, bone mineral density; BMI, body mass index; eGFR, estimated glomerular filtration rate; HBV, hepatitis B virus; HCV, hepatitis C virus; IGF-1, insulin-like growth factor 1; INR, international normalized ratio; M2BPGi, Mac-2 binding protein glycosylation isomer; PBC, primary biliary cholangitis.

(n = 46), hepatitis C virus (n = 99), alcohol (n = 63), primary biliary cholangitis (n = 62), and others (n = 54), which included autoimmune hepatitis, nonalcoholic fatty liver disease, and cryptogenic CLD. The median pentosidine level was 0.0598 (0.0465–0.0878) μg/mL. The median values of BMD at the lumbar spine, femoral neck, and total hip were 1.07 (0.90–1.21) g/cm², 0.76 (0.66–0.88) g/cm², and 0.83 (0.71–0.94) g/cm², respectively. The prevalence of osteoporosis was 31.8% (103/324). S1 Table summarizes the baseline characteristics across etiologies, and the following variables significantly differed among the etiology groups: gender, age, BMI, prevalence of diabetes and current smoking, total bilirubin, albumin, PT-INR, M2BPGi, IGF-1, pentosidine, and all BMDs. Notably, patients with ALD had the highest prevalence of LC among the five groups [96.8% (61/63), *p* < 0.001; adjusted residual = |7.0|].

## Comparison of clinical characteristics between patients with and without prevalent fractures

As shown in Table 1, 105 (32.4%) patients had prevalent fractures in the following locations: vertebra, n = 85; distal radius, n = 14; proximal femur, n = 10; rib, n = 10; pelvis, n = 6;

**Table 2. Significant factors associated with prevalent fractures in patients with chronic liver disease.**

| Variable | Univariate | | Multivariate | |
|---|---|---|---|---|
| | OR (95%CI) | *p* value | OR (95%CI) | *p* value |
| Age (years) | 1.084 (1.056–1.113) | < 0.001 | 1.073 (1.042–1.106) | < 0.001 |
| BMI (kg/m$^2$) | 0.933 (0.878–0.993) | < 0.001 | | |
| Diabetes mellitus | 1.111 (0.657–1.877) | 0.695 | | |
| Chronic kidney disease | 1.638 (1.024–2.620) | 0.039 | | |
| Liver cirrhosis | 1.937 (1.187–3.163) | 0.008 | | |
| eGFR (mL/min/1.73m$^2$) | 0.982 (0.969–0.994) | 0.005 | | |
| IGF-1 (ng/mL) | 0.984 (0.976–0.993) | < 0.001 | | |
| Pentosidine (x10$^2$) (μg/mL) | 1.038 (1.007–1.068) | 0.014 | 1.069 (1.032–1.107) | < 0.001 |
| Lumbar spine BMD (g/cm$^2$) | 0.071 (0.022–0.232) | < 0.001 | | |
| Femoral neck BMD (g/cm$^2$) | 0.002 (0.000–0.014) | < 0.001 | | |
| Total hip BMD (g/cm$^2$) | 0.002 (0.000–0.013) | < 0.001 | 0.006 (0.001–0.046) | < 0.001 |
| Osteoporosis | 5.101 (3.070–8.477) | < 0.001 | | |

BMD, bone mineral density; BMI, body mass index; CI, confidence interval; eGFR, estimated glomerular filtration rate; IGF-1, insulin-like growth factor 1; OR, odds ratio.

proximal humerus, n = 4; and lower extremity, n = 2. There were no patients with fractures at the time of study entry. Patients with prevalent fractures were significantly older (*p* < 0.001) and had a lower body mass index (BMI; *p* = 0.012) and higher prevalence of CKD (50.5% vs. 38.4%; *p* = 0.039), and LC (68.6% vs. 53.0%; *p* = 0.008) than those without prevalent fractures. Patients in the fracture group had significantly lower eGFRs (*p* = 0.012) and IGF-1 levels (*p* = 0.001) and higher M2BPGi (*p* = 0.003) and pentosidine (*p* = 0.004) levels than those in the non-fracture group. The BMD values of the lumbar spine, femoral neck, and total hip were significantly lower in the fracture group than in the non-fracture group (*p* < 0.001 for all). The prevalence of osteoporosis was significantly higher in the fracture group than in the non-fracture group (56.2% vs. 20.1%; *p* < 0.001).

## Significant factors associated with prevalent fractures

The univariate analysis identified the following eleven variables that were significantly related to prevalent fractures: age, BMI, CKD, LC, eGFR, IGF-1, pentosidine, BMD of the lumbar spine, femoral neck, and total hip, and osteoporosis (S2 Table). Finally, the following three variables were retained as independent factors associated with prevalent fractures: older age [OR = 1.073, 95% confidence interval (CI) = 1.042–1.106, *p* < 0.001], higher pentosidine levels (OR = 1.069, 95%CI = 1.032–1.107, *p* < 0.001), and lower BMD of the total hip (OR = 0.006, 95%, CI = 0.001–0.046, *p* < 0.001) (Table 2).

## Clinical characteristics based on the baseline pentosidine levels

Patients were classified into three groups according to the baseline plasma pentosidine levels, as described in the Methods. The prevalence of L-Pen, I-Pen, and H-Pen was 25.0% (81/324), 50.6% (164/324), and 24.4% (79/324), respectively (Table 3). The H-Pen group showed significantly worse liver functional reserve (total bilirubin, albumin, and PT-INR) compared with the other two groups (*p* < 0.001 for all; Table 3; Fig 1A–1C). In contrast, the L-Pen group showed significantly better kidney function (creatinine and eGFR) compared with the other two groups (Table 3; Fig 1D and 1E). The H-pen group showed significantly higher levels of M2BPGi and the highest prevalence of LC [88.6% (70/79), *p* < 0.001; adjusted residual = |6.3|],

**Table 3. Characteristics of the three groups classified according to the plasma pentosidine levels.**

| Variable | L-Pen | I-Pen | H-Pen | p value |
|---|---|---|---|---|
| Patients, n (%) | 81 (25.0) | 164 (50.6) | 79 (24.4) | |
| Man, n (%) | 31 (38.3) | 83 (50.6) | 45 (57.0) | 0.052 |
| Age (years) | 68.0 (54.5–72.5) | 71.0 (61.0–77.0) | 69.0 (55.0–77.0) | 0.004 |
| BMI (kg/m$^2$) | 24.3 (21.1–27.5) | 23.1 (21.1–25.9) | 22.3 (19.9–25.5) | 0.039 |
| Current smoking, n (%) | 19 (23.5) | 41 (25.0) | 26 (32.9) | 0.326 |
| Current drinking, n (%) | 3 (3.7) | 15 (9.1) | 18 (22.8) | < 0.001 |
| Menopause, n (%) | 46 (92.0) | 77 (95.1) | 32 (94.1) | 0.774 |
| Diabetes mellitus, n (%) | 19 (23.5) | 44 (26.8) | 22 (27.8) | 0.795 |
| Chronic kidney disease, n (%) | 23 (28.4) | 73 (44.5) | 41 (51.9) | 0.008 |
| Liver cirrhosis, n (%) | 26 (32.1) | 92 (56.1) | 70 (88.6) | < 0.001 |
| Child-Pugh B+C, n (%) | 4 (15.4) | 22 (23.9) | 51 (72.9) | < 0.001 |
| Etiology | | | | |
| HBV/HCV/Alcohol/PBC/other, n | 19/21/7/19/15 | 21/53/29/30/31 | 6/25/27/13/8 | 0.001 |
| Total bilirubin (mg/dL) | 0.6 (0.5–0.9) | 0.7 (0.5–1.0) | 1.1 (0.6–2.1) | < 0.001 |
| Albumin (g/dL) | 4.1 (3.9–4.4) | 4.0 (3.7–4.3) | 3.4 (2.8–3.8) | < 0.001 |
| Prothrombin time INR | 1.00 (0.96–1.07) | 1.04 (0.96–1.13) | 1.18 (1.06–1.35) | < 0.001 |
| Creatinine (mg/dL) | 0.7 (0.6–0.9) | 0.8 (0.7–1.0) | 0.9 (0.7–1.2) | < 0.001 |
| eGFR (mL/min/1.73m$^2$) | 66 (59–81) | 63 (51–76) | 59 (42–77) | 0.007 |
| M2BPGi (C.O.I) | 0.91 (0.65–1.54) | 1.66 (0.96–3.36) | 6.19 (2.71–8.60) | < 0.001 |
| IGF-1 (ng/mL) | 81 (65–106) | 64 (45–82) | 47 (32–61) | < 0.001 |
| 25(OH)D (ng/mL) | 14.4 (11.1–17.9) | 14.0 (9.7–18.3) | 11.1 (9.0–14.7) | 0.064 |
| Vitamin D deficiency, n (%) | 69 (85.2) | 142 (87.1) | 69 (88.5) | 0.826 |
| Lumbar spine BMD (g/cm$^2$) | 1.06 (0.90–1.24) | 1.10 (0.90–1.22) | 1.04 (0.88–1.18) | 0.366 |
| Femoral neck BMD (g/cm$^2$) | 0.75 (0.68–0.90) | 0.76 (0.66–0.88) | 0.78 (0.65–0.87) | 0.806 |
| Total hip BMD (g/cm$^2$) | 0.84 (0.72–0.96) | 0.82 (0.71–0.93) | 0.82 (0.68–0.92) | 0.395 |
| Osteoporosis, n (%) | 23 (28.4) | 52 (31.7) | 28 (35.4) | 0.632 |
| Prevalent fracture, n (%) | 17 (21.0) | 53 (32.3) | 35 (44.3) | 0.007 |

Values are presented as median (interquartile ranges) or relative frequencies (percentages). Statistical analysis was performed using the chi-squared test or the Kruskal-Wallis test, as appropriate. 25(OH)D, 25-hydroxyvitamin D; AL, alcohol; BMD, bone mineral density; BMI, body mass index; eGFR, estimated glomerular filtration rate; HBV, hepatitis B virus; HCV, hepatitis C virus; IGF-1, insulin-like growth factor 1; INR, international normalized ratio; Mac-2 binding protein glycosylation isomer; PBC, primary biliary cholangitis.

CKD [51.9% (41/79), $p = 0.008$; adjusted residual = |2.0|], and ALD [34.2% (27/79), $p = 0.001$; adjusted residual = |3.8|], whereas the L-Pen group had significantly lower levels of M2BPGi and the lowest prevalence of LC [32.1% (26/81), $p < 0.001$; adjusted residual = |5.5|], CKD [28.4% (23/81), $p = 0.008$; adjusted residual = |2.9|], and ALD [8.6% (7/81), $p = 0.001$; adjusted residual = |2.8|] (Table 3; Fig 1F–1H). Notably, the H-Pen group had the highest prevalence of prevalent fractures [44.3% (35/79), $p = 0.007$; adjusted residual = |2.6|], whereas the L-Pen group showed the lowest prevalence [21.0% (17/81), $p = 0.007$; adjusted residual = |2.5|] (Table 3; Fig 1I). The prevalence of LC ($p < 0.001$), CKD ($p = 0.003$), and prevalent fractures ($p = 0.002$) significantly increased in a stepwise manner with elevation in pentosidine levels (Fig 1G–1I).

## Optimal cutoff value of plasma pentosidine for predicting prevalent fractures

We performed an ROC curve analysis to determine the optimal cutoff value of plasma pentosidine for predicting the presence or absence of prevalent fractures. The area under the ROC

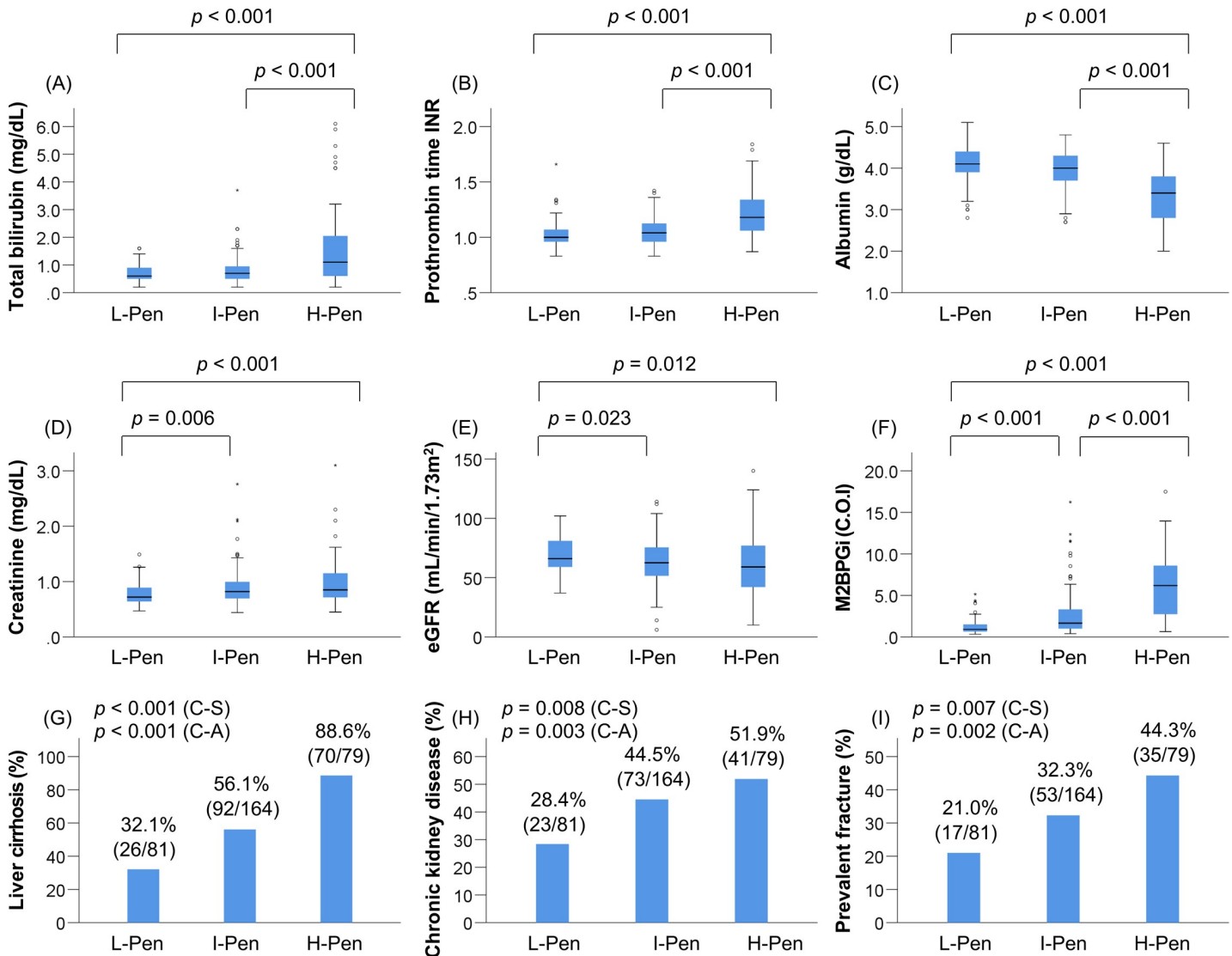

**Fig 1. Comparison of clinical characteristics among the low (L)-pentosidine (Pen), intermediate (I)-Pen, and high (H)-Pen groups.** The (A) total bilirubin levels and (B) prothrombin time-international normalized ratio were significantly higher in the H-Pen group than in the L-Pen and I-Pen groups. (C) The levels of albumin were significantly lower in the H-Pen group than in the L-Pen and I-Pen groups. The (D) creatinine levels were significantly higher and (E) estimated glomerular filtration rate was significantly lower in the I-Pen and H-Pen groups than in the L-Pen group. (F) The levels of mac-2 binding protein glycosylation isomer were highest among the H-Pen group. (G) (H) (I) The H-Pen group had the highest prevalence of liver cirrhosis (chi-squared test: $p < 0.001$), chronic kidney disease (chi-squared test: $p = 0.008$), and prevalent fractures (chi-squared test: $p = 0.007$) among the three groups. C-A, Cochran–Armitage trend test; C-S, chi-squared test.

curve (AUC), optimal cutoff value, sensitivity, and specificity were 0.60, 0.0545 μg/mL, 0.714, and 0.452, respectively (S2 Fig). These results suggest that the plasma pentosidine levels are not very useful in predicting prevalent fractures in the present study.

## Relationship between plasma pentosidine levels and liver functional reserve

As described above, the H-Pen group had significantly worse liver functional reserve (total bilirubin, albumin, and PT-INR) and the highest prevalence of LC and CKD. We also identified the factors that significantly and independently correlated with plasma pentosidine levels. Plasma pentosidine levels increased stepwise as the liver disease progressed (Fig 2A), and

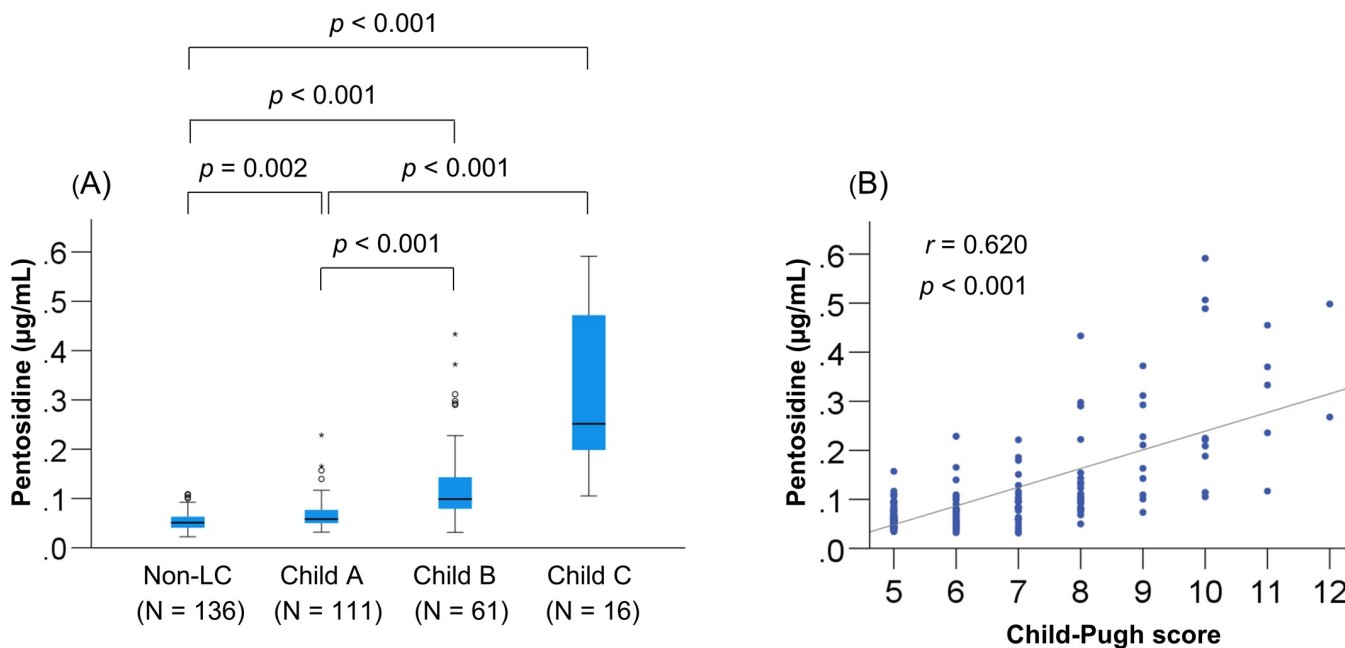

**Fig 2. Relationship between plasma pentosidine levels and Child-Pugh class.** (A) The plasma pentosidine levels were significantly higher in patients with Child-Pugh class B and C [decompensated liver cirrhosis (LC)] than in those with non-LC and Child-Pugh class A (compensated LC), and (B) significantly correlated with Child-Pugh scores in patients with LC.

significantly correlated with Child-Pugh scores (Fig 2B). The correlation between plasma pentosidine levels and baseline clinical characteristics was investigated using the Spearman's rank correlation test. Baseline factors that significantly correlated with plasma pentosidine levels were as follows: BMI, liver functional reserve measurements (total bilirubin, albumin, and PT-INR), creatinine, eGFR, M2BPGi, IGF-1, and 25(OH)D (S3 Table). In the multiple regression analysis, the following six variables were significantly and independently related to plasma pentosidine levels (S4 Table): BMI ($p = 0.015$), total bilirubin ($p < 0.001$), albumin ($p < 0.001$), PT-INR ($p = 0.001$), creatinine ($p = 0.001$), and prevalent fracture ($p = 0.015$). Taken together, liver functional reserve factors (total bilirubin, albumin, and PT-INR) were significantly and independently associated with plasma pentosidine levels in patients with CLD. Intriguingly, these factors are components of the Child–Pugh scoring system.

### Comparison of clinical characteristics among patients with and without high pentosidine levels and/or osteoporosis

The 324 patients were stratified into four groups based on a combination of high/non-high pentosidine and osteoporosis/non-osteoporosis, as follows: (1) patients without high pentosidine levels or osteoporosis (170/324; 52.5%); (2) patients with high pentosidine levels alone (51/324; 15.7%); (3) patients with osteoporosis alone (75/324; 23.1%); and (4) patients with both high pentosidine levels and osteoporosis (28/324; 8.6%) (S5 Table). Gender, age, BMI, prevalence of LC, liver functional reserve measurements (total bilirubin, albumin, and PT-INR), creatinine, M2BPGi, IGF-1, 25(OH)D, and pentosidine significantly differed among the groups. Notably, the prevalence of prevalent fractures was highest in patients with both high pentosidine levels and osteoporosis [75.0% (21/28), $p < 0.001$; adjusted residual = |5.0|; Fig 3], whereas the prevalence was lowest in patients without high pentosidine levels or osteoporosis [18.8% (32/170), $p < 0.001$; adjusted residual = |5.5|; Cramér's V = 0.390]. The

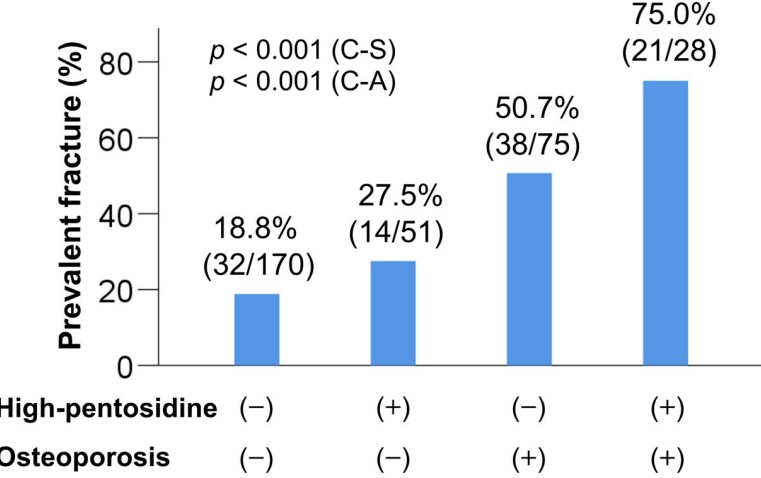

**Fig 3. Comparison of the prevalence of prevalent fractures among four groups.** (1) the osteoporosis (−)/high pentosidine levels (−) group, (2) the osteoporosis (−) / high pentosidine (+) group, (3) the osteoporosis (+)/high pentosidine (−) group, and (4) the osteoporosis (+) /high pentosidine (+) group. The prevalence of prevalent fractures was significantly highest in the osteoporosis (+) /high pentosidine (+) group (chi-squared test: $p < 0.001$). The prevalence of prevalent fractures significantly increased stepwise with complications of high pentosidine levels and/or osteoporosis (Cochran–Armitage trend test: $p < 0.001$). C-A, Cochran–Armitage trend test; C-S, chi-squared test.

prevalence of prevalent fractures significantly increased in a stepwise manner with high pentosidine levels and the presence of osteoporosis.

## Discussion

A decrease in BMD (osteoporosis) and impaired bone quality are associated with bone fragility and consequent fractures [12, 13]. However, bone quality, in term of tissue material properties, has not been investigated in patients with CLD. Among several risk factors for fractures, prevalent fractures are especially significant in predicting the occurrence of further fragility fractures [17]. Therefore, the present study focused on bone quality and aimed to clarify its involvement in prevalent fractures among patients with CLD.

Non-enzymatic collagen cross-links (AGEs), for which pentosidine is a surrogate biomarker, impair bone mechanical properties and cause bone fragility [12, 13]. Previous studies in postmenopausal women and patients with diabetes revealed that higher levels of urinary and serum pentosidine were significantly and independently related to the prevalence of fractures [17–20]. In the present study, we demonstrated that higher plasma pentosidine levels, older age, and lower total hip BMD were significantly and independently associated with prevalent fractures. Therefore, we classified the patients into three groups according to baseline plasma pentosidine levels and investigated prevalent fractures. Notably, the H-Pen group had the highest prevalence of prevalent fractures, whereas the L-Pen group had the lowest prevalence. In addition, the rate of prevalent fractures was higher in patients with both osteoporosis and high pentosidine levels compared with patients with no osteoporosis and intermediate or low pentosidine levels and patients with high pentosidine levels or osteoporosis alone. These results suggest that both higher plasma pentosidine levels (impaired bone quality) and lower BMD are cooperatively associated with prevalent fracture in patients with CLD.

To the best of our knowledge, this is the first report to evaluate plasma pentosidine levels in patients with CLD. Pentosidine, a fluorescent intermolecular cross-linking type AGE, is induced by glycation and/or oxidation and increases with age. Pentosidine production is also elevated in several diseases, including chronic kidney dysfunction and diabetes [12, 13]. A

previous report demonstrated that plasma pentosidine levels significantly and linearly correlate with cortical bone pentosidine levels [29]. Thus, plasma pentosidine is a potential surrogate biomarker for bone quality. In the present study, plasma pentosidine levels increased stepwise as the disease stage progressed (from non-LC to Child–Pugh classes A, B, and C) and significantly correlated with Child–Pugh scores. Specifically, patients with decompensated LC had remarkably higher plasma pentosidine levels. Therefore, plasma pentosidine could also predict the disease stage or estimate the liver functional reserve. Moreover, the liver functional reserve factors (total bilirubin, albumin, and PT-INR, which are also components of the Child-Pugh scoring system), creatinine, and prevalent fractures (but not patient age) were significantly and independently associated with plasma pentosidine levels in patients with CLD. In general, as the liver disease stage advances, the kidney function worsens in patients with CLD [30]. Therefore, CKD associated with CLD could coordinately or synergistically elevate plasma pentosidine levels. Similarly, serum pentosidine levels in patients with rheumatoid arthritis (RA) are significantly higher compared with those in healthy individuals [31]; serum pentosidine levels correlate with age in healthy individuals, but not in patients with RA. Additionally, serum pentosidine levels positively correlate with the levels of inflammatory markers, such as C-reactive protein, erythrocyte sedimentation, and interleukin (IL)-6, in patients with RA. In patients with CLD, the production of reactive oxygen species and inflammatory cytokines, such as IL-6, increases as chronic hepatitis progresses [32, 33]. These results suggest that higher pentosidine levels are associated with advanced liver disease and chronic inflammatory conditions rather than age. Therefore, careful attention for fractures should be given to patients with CLD, especially in those with advanced disease, irrespective of age.

In a previous study, administration of selective estrogen receptor modulators ameliorated detrimental collagen cross-linking and the consequent loss of bone strength in rabbits with ovariectomy [34]. Similarly, treatment with parathyroid hormone (1–34) induced enzymatic collagen cross-links, increased bone volume, and decreased pentosidine (non-enzymatic cross-links) levels in monkeys with ovariectomy, thereby improving bone strength [35]. We previously reported that administration of denosumab, a human monoclonal antibody against the receptor activator of nuclear factor kappa-B ligand, increased BMD, suppressed bone turnover, and decreased plasma pentosidine in CLD patients with osteoporosis [36]. The current clinical practice guidelines on CLD recommend supplementation with calcium and vitamin D for a T-score < −1.5 and administration of bisphosphonates to improve BMD in CLD patients with osteoporosis [37]. In the future, customized osteoporosis treatment strategies, including improved "bone quality", as well as BMD, should be considered to prevent fractures in these patients.

This study has some limitations. First, this study did not include healthy controls. Second, this was a cross-sectional study and, thus, did not prospectively assess the relationship between plasma pentosidine levels and the occurrence of fractures. Finally, we did not exclude patients with renal dysfunction and/or diabetes, who tend to have elevated plasma pentosidine levels, given that patients with CLD are frequently complicated by these disorders.

## Conclusions

In the present study, we demonstrated that higher plasma pentosidine levels were significantly and independently associated with prevalent fractures in patients with CLD. High plasma pentosidine levels closely associated with factors related to advanced disease. Pentosidine may be useful for predicting fracture risk and should be closely monitor in CLD patients with advanced disease. Comprehensive assessment (including plasma pentosidine levels) and

customized treatment strategies for osteoporosis and bone quality are essential to prevent fractures in patients with CLD, especially in those with advanced liver disease.

## Supporting information

**S1 Fig. Classification based on the baseline plasma pentosidine levels.** The median (interquartile range) pentosidine level was 0.0598 (0.0465–0.0886) μg/mL. The 324 patients were divided into three groups: (1) the low pentosidine (L-Pen) group had pentosidine levels ≤0.0465 μg/mL (first quartile); (2) the intermediate pentosidine group had pentosidine levels 0.0465–0.0886 μg/mL (third quartile); and (3) the high pentosidine group had pentosidine levels ≥0.0886 μg/mL.
(TIF)

**S2 Fig. The receiver operating characteristic (ROC) curve analysis of plasma pentosidine for predicting prevalent fractures.** The plasma pentosidine cutoff value was 0.0545 μg/mL with area under the ROC curve (AUC), specificity, and sensitivity of 0.60, 0.714, and 0.452, respectively.
(TIF)

**S1 Table. Comparison of baseline characteristics across etiologies.**
(DOCX)

**S2 Table. Univariate analysis of factors associated with prevalent fractures.**
(DOCX)

**S3 Table. Correlation between plasma pentosidine levels and baseline characteristics.**
(DOCX)

**S4 Table. Multiple regression analysis of factors associated with plasma pentosidine levels.**
(DOCX)

**S5 Table. Baseline characteristics of patients with and without high pentosidine levels and/ or osteoporosis.**
(DOCX)

## Acknowledgments

We thank the medical staff at Fuji City General Hospital who were involved in the data collection.

## Author Contributions

**Conceptualization:** Chisato Saeki, Mitsuru Saito.

**Data curation:** Chisato Saeki, Tomoya Kanai, Masanori Nakano, Yuichi Torisu.

**Formal analysis:** Chisato Saeki, Akihito Tsubota.

**Supervision:** Mitsuru Saito, Tsunekazu Oikawa, Masayuki Saruta, Akihito Tsubota.

**Writing – original draft:** Chisato Saeki.

**Writing – review & editing:** Chisato Saeki, Mitsuru Saito, Akihito Tsubota.

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
