## [Decision Letter · Decision Letter 0]

26 Jan 2021

PONE-D-21-00527

Plasma pentosidine levels are associated with prevalent fractures in patients with chronic liver disease

PLOS ONE

Dear Dr. Saeki,

Thank you for submitting your manuscript to PLOS ONE. After careful consideration, we feel that it has merit but does not fully meet PLOS ONE’s publication criteria as it currently stands. Therefore, we invite you to submit a revised version of the manuscript that addresses the points raised during the review process.

While both reviewers found your study potentially interesting, there are also important issues that need to be considered/disclosed prior to publication (heterogeneity of the cohort, a somewhat superficial characterization of the non-cirrhotic participants etc.). Because of that, the bar for the revision will be relatively high. Among others, the etiology of liver fibrosis might influence the rate of fractures and its role should be carefuly analysed. It should be also considered for the multivariate model.

We look forward to receiving your revised manuscript.

Kind regards,

Pavel Strnad

Academic Editor

PLOS ONE

Journal Requirements:

2. In your Methods section, please provide additional information about the participant recruitment method and the demographic details of your participants. Please ensure you have provided sufficient details to replicate the analyses such as:

a) a statement as to whether your sample can be considered representative of a larger population, and

b) a description of how participants were recruited.

3.PLOS ONE requires experimental methods to be described in enough detail to allow suitably skilled investigators to fully replicate and evaluate your study. See https://journals.plos.org/plosone/s/submission-guidelines#loc-materials-and-methods for more information.

To comply with PLOS ONE submission guidelines, in your Methods section, please provide a more detailed description of your methodology in your Biochemical assessment section. Please ensure that you describe the sources and catalog numbers of all ELISA assays, antibodies, etc. in the methods section of your manuscript. For antibodies, please also include the dilutions used in your experiments.

Reviewers' comments:

Reviewer's Responses to Questions

**Comments to the Author**

1. Is the manuscript technically sound, and do the data support the conclusions?

Reviewer #1: Partly

Reviewer #2: Partly

2. Has the statistical analysis been performed appropriately and rigorously? 

Reviewer #1: No

Reviewer #2: Yes

3. Have the authors made all data underlying the findings in their manuscript fully available?

Reviewer #1: Yes

Reviewer #2: Yes

4. Is the manuscript presented in an intelligible fashion and written in standard English?

Reviewer #1: Yes

Reviewer #2: Yes

5. Review Comments to the Author

Reviewer #1: In their study the authors determined the relationship between plasma levels of pentosidine, which represents a surrogate marker for advanced glycation end products (AGEs), and prevalent fractures in 324 patients with chronic liver disease. They obtained information on prevalent fractures through medical interviews, records, and/or radiography. After classifying the patients into three groups according to their pentosidine serum levels, they found out that the high pentosidine-group had the highest prevalence of liver cirrhosis and prevalent fractures, whereas the low pentosidine group showed the lowest prevalence of fractures and liver cirrhosis. They were able to show that pentosidine serum levels significantly correlate with liver functional reserve factors and a hepatic fibrosis marker, but not with age.

The authors concluded their study by stating that pentosidine is a significant independent factor related to prevalent fractures in patients with CLD and should be closely followed in individuals with advanced liver disease.

Despite these novel results, the manuscript has a couple of weaknesses that greatly diminish the value of the presented data.

Major weaknesses:

1. As explained by the authors themselves, pentosidine production is increased in several other diseases, such as chronic kidney disease or diabetes. However, patients displaying these diagnoses were not excluded or taken into account in the statistical analyses, and thus results might be biased. These confounders need to be included into a multivariable analysis. Alternatively, subgroup-analyses can be performed.

2. The authors excluded individuals with pathological processes and prolonged glucocorticoid administration. Similar to 1) a further improvement might be achieved by including other confounders or risk factor for osteoporosis, such as pathological alcohol consumption or excessive smoking into the statistical analyses. How are the different liver disease etiologies taken into account (alcoholic liver cirrhosis etc.)?

3. Please elaborate more on the clinical consequences. What is the AUROC of pentosidine for the prediction of fractures in CLD patients? Are there any data showing whether high pentosidine serum levels associated with increased mortality in these patients?

Minor comments:

- Table 1: I would prefer listing “diabetes mellitus” above “liver cirrhosis”, under “BMI”. This would place “liver cirrhosis” and “etiology” below each other as they belong together.

- Table 2: Please notice different style of “P-value” and “p value” (page 12, line 182).

- Improvement suggestion instead of “as medians (interquartile ranges) and numbers (percentages)” better “as medians (interquartile ranges) and relative frequencies (%)” (page 8, line 126; page 10, line 151).

Reviewer #2: Dear Dr. Saeki, dear Co-authors,

I have read your submission with great interest.

Unfortunately I find the study to have multiple limitations, that greatly diminish its value:

1. The cohort investigated lacks sufficient characterisation with regard to “CLDs”. This produces a high amount of confounders. What diseases were considered chronic, especially in patients without cirrhosis? For example, was any given amount of alcohol consumption regarded as a chronic liver disease or were only diseases included, that had already provided liver injury in substantial amounts? How was such liver injury measured (did the authors use any scoring systems, diagnostic tools like transient elastography or histology)? The way the information is provided, I can see too many confounders disturbing the investigated results. A smaller, but sufficiently characterised cohort might have achieved less confounding bias.

2. Comprehensibly, the authors did not exclude patients with chronically impaired renal function, as it represents a common (co)morbidity in liver disease. This has been vocally addressed, but still confounds pentosidine plasma levels independent of CLD.

3. Exclusion criteria mention patients previously treated with GC for 3 months, but do not provide information on or exclude patients with other reasons for osteoporosis, such as postmenopausal women with or without hormone supplementation or patients with other reasons for a substantial vitamin d deficiency. Additionally, did the authors include history of fractures before the primary diagnosis of a liver disease? How did the authors manage patients with CLD and fractures after the age of 40 years that were induced through traumatic injury?

4. Cirrhosis is not sufficiently defined, as scoring systems and diagnostic information are lacking (Child-Pugh, MELD, …). A dichotomous comparison between cirrhotic and non-cirrhotic patients would have been of great interest, as the authors central question for pentosidine measurement involves the differences between sub cohorts of CLD.

5. A comparison of the different CLD aetiologies would have been interesting. Here, especially the univariate analysis of significant factors associated with prevalent fractures lacks inclusion of CLD aetiologies. A confounding bias is likely.

6. The characterisation of pentosidine groups has raised important questions, that have not been sufficiently attended. For example, did plasma pentosidine levels correlate with Child-Pugh or MELD in patients with liver cirrhosis? This information would be especially helpful, as laboratory parameters in the different pentosidine groups differed significantly.

6. PLOS authors have the option to publish the peer review history of their article (what does this mean?). If published, this will include your full peer review and any attached files.

Reviewer #1: No

Reviewer #2: No

---

## [Author Response · Author response to Decision Letter 0]

15 Feb 2021

RESPONSES TO THE REVIEWER

We wish to express our appreciation to the reviewers for the critical and insightful comments on our manuscript. We feel the reviewers’ comments have helped us markedly improve our manuscript. Our point-by-point responses to the reviewers’ comments are listed below.

Review Comments to the Author:

Reviewer #1:

In their study the authors determined the relationship between plasma levels of pentosidine, which represents a surrogate marker for advanced glycation end products (AGEs), and prevalent fractures in324 patients with chronic liver disease. They obtained information on prevalent fractures through medical interviews, records, and/or radiography. After classifying the patients into three groups according to their pentosidine serum levels, they found out that the high pentosidine-group had the highest prevalence of liver cirrhosis and prevalent fractures, whereas the low pentosidine group showed the lowest prevalence of fractures and liver cirrhosis. They were able to show that pentosidine serum levels significantly correlate with liver functional reserve factors and a hepatic fibrosis marker, but not with age. The authors concluded their study by stating that pentosidine is a significant independent factor related to prevalent fractures in patients with CLD and should be closely followed in individuals with advanced liver disease. Despite these novel results, the manuscript has a couple of weaknesses that greatly diminish the value of the presented data.

Major weaknesses:

(1) As explained by the authors themselves, pentosidine production is increased in several other diseases, such as chronic kidney disease or diabetes. However, patients displaying these diagnoses were not excluded or taken into account in the statistical analyses, and thus results might be biased. These confounders need to be included into a multivariable analysis. Alternatively, subgroup-analyses can be performed.

Responses: We appreciate the reviewer’s critical comments. Certainly, pentosidine production increases in patients with CKD and diabetes. As indicated by the reviewer, we did not exclude patients with CKD/diabetes because patients with CLD (especially with advanced liver disease) are frequently complicated by these disorders. We provide a new supplementary figure only for review (R1 Figure), which shows the prevalence of CKD/diabetes (Non-LC vs. LC) in this study cohort. We newly added the prevalence of CKD to the revised Table 1. As suggested by the reviewer, we performed univariate and multiple logistic regression analyses that included new variables (such as CKD and diabetes) to identify significant and independent factors related to prevalent fractures. In the univariate analysis, CKD (but not diabetes) was associated with prevalent fractures (p = 0.039; revised S2 Table). Finally, neither CKD nor diabetes were significant and independent in the multivariate analysis (revised Table 2). In addition, we newly performed multiple regression analysis to identify significant and independent factors affecting plasma pentosidine levels (new S4 Table). As a result, BMI, total bilirubin, albumin, prothrombin time (PT) INR, creatinine, and prevalent fractures were significantly and independently associated with plasma pentosidine levels. These results suggest that liver functional reserve factors (total bilirubin, albumin, and PT-INR) are significant independent factors affecting the plasma pentosidine levels, even when potential cofounders (such as CKD and diabetes) were taken into consideration by using multivariate analysis.

(2) The authors excluded individuals with pathological processes and prolonged glucocorticoid administration. Similar to 1) a further improvement might be achieved by including other confounders or risk factor for osteoporosis, such as pathological alcohol consumption or excessive smoking into the statistical analyses. How are the different liver disease etiologies taken into account (alcoholic liver cirrhosis etc.)?

Responses: We thank the reviewer for the above important points. Smoking and heavy alcohol consumption are known as risk factors for osteoporosis and osteoporotic fractures. The Fracture Risk Assessment tool (FRAX) algorism developed by the World Health Organization (WHO) to evaluate the 10-year probability of osteoporotic fracture includes components of current smoking and heavy alcohol consumption (>3 units/day) (Kanis JA, et al. Osteoporos Int. 2008). In the revised manuscript, we took possible cofounders or risk factors [such as current smoking, current drinking (>3 units/day), and CLD etiology including alcoholic liver disease] into consideration. However, statistical analyses indicated that they did not significantly differ between the fracture and non-fracture groups (revised Table 1) and were not significantly associated with prevalent fractures (revised S2 Table). Thus, current smoking, current drinking, and CLD etiology were not significant factors related to prevalent fractures in this study cohort.

(3) Please elaborate more on the clinical consequences. What is the AUROC of pentosidine for the prediction of fractures in CLD patients? Are there any data showing whether high pentosidine serum levels associated with increased mortality in these patients?

Responses: As suggested by the reviewer, we performed an ROC curve analysis to determine the optimal cutoff value of plasma pentosidine for predicting prevalent fractures [R2 Figure (only for review)]: the optimal cutoff value, AUC, sensitivity, and specificity were 0.0545 μg/mL, 0.60, 0.714, and 0.452, respectively. These results suggest that the plasma pentosidine levels may not be very useful in predicting prevalent fractures.

Our study demonstrated that the plasma pentosidine levels significantly correlated with liver functional reserve factors (total bilirubin, albumin, and PT-INR) and M2BPGi (hepatic fibrosis marker). Additionally, we newly compared the plasma pentosidine levels among patients with non-LC and LC (stratified by the Child-Pugh classification; new Figure 2A): patients with LC, especially with decompensated LC (Child-Pugh B/C), had significantly higher levels of plasma pentosidine than those with compensated LC (Child-Pugh A) and non-LC. Furthermore, we investigated the correlation between plasma pentosidine levels and Child-Pugh scores in patients with LC (new Figure 2B): plasma pentosidine levels significantly correlated with Child-Pugh scores. Thus, we believe that plasma pentosidine levels could be a surrogate biomarker for poor prognosis in patients with CLD. However, the observational period was not long enough to analyze the prognosis in this study. In the future, we are willing to investigate and report the relationship between plasma pentosidine levels and prognosis in patients with CLD.

Minor comments:

- Table 1: I would prefer listing “diabetes mellitus” above “liver cirrhosis”, under “BMI”. This would place “liver cirrhosis” and “etiology” below each other as they belong together.

- Table 2: Please notice different style of “P-value” and “p value” (page 12, line 182).

- Improvement suggestion instead of “as medians (interquartile ranges) and numbers (percentages)” better “as medians (interquartile ranges) and relative frequencies (%)” (page 8, line 126; page 10, line 151).

Responses: We appreciate the kindly reviewer’s suggestions for improvement. As instructed by the reviewer, we revised the original Table 1 and Table 2 appropriately (please see the revised Table 1 and Table 2).

Reviewer #2:

I have read your submission with great interest. Unfortunately I find the study to have multiple limitations, that greatly diminish its value:

(1) The cohort investigated lacks sufficient characterisation with regard to “CLDs”. This produces a high amount of confounders. What diseases were considered chronic, especially in patients without cirrhosis? For example, was any given amount of alcohol consumption regarded as a chronic liver disease or were only diseases included, that had already provided liver injury in substantial amounts? How was such liver injury measured (did the authors use any scoring systems, diagnostic tools like transient elastography or histology)? The way the information is provided, I can see too many confounders disturbing the investigated results. A smaller, but sufficiently characterised cohort might have achieved less confounding bias.

Responses: We are thankful for the reviewer’s critical suggestions. We newly described our responses in the Method section (lines 99–114). Chronic liver disease (CLD), including hepatitis B or C, alcoholic liver disease (ALD), autoimmune hepatitis, primary biliary cholangitis, non-alcoholic fatty liver disease, and cryptogenic hepatitis, was defined as persistent liver damage characterized by abnormal laboratory tests (such as elevated liver enzymes possibly due to each etiology) that lasted for at least 6 months and/or histopathological findings on liver biopsy specimens. ALD was diagnosed based on CLD with current and/or past history of heavy alcohol consumption (>3 units/day) and without other etiologies. We estimate hepatic fibrosis by using transient elastography in clinical practice, however, not all patients with CLD underwent elastography as routine examination.

We newly investigated the characteristics of each etiology, as shown in new S1 Table. In particular, patients with ALD were younger and had a higher prevalence of LC and consequent worse liver functional reserve/fibrosis marker (total bilirubin, albumin, PT-INR, and M2BPGi), compared to those with non-ALD. However, the univariate analysis showed that etiology was not significantly associated with prevalent fractures in our study (revised S2 Table). In addition, the multiple regression analysis revealed that etiology was not significantly and independently associated with plasma pentosidine levels (new S4 Table).

(2) Comprehensibly, the authors did not exclude patients with chronically impaired renal function, as it represents a common (co)morbidity in liver disease. This has been vocally addressed, but still confounds

pentosidine plasma levels independent of CLD.

Responses: We appreciate the reviewer’s critical comments. Pentosidine production is likely to increase in patients with CKD and diabetes. However, we did not exclude patients with CKD/diabetes because patients with CLD (especially with advanced liver disease) are frequently complicated by these disorders [revised Table 1,R1 Figure (only for review)]. Therefore, we performed multiple regression analysis to identify significant and independent factors related to the plasma pentosidine levels (new S4 Table). As a result, BMI, total bilirubin, albumin, prothrombin time (PT) INR, creatinine, and prevalent fractures were significantly and independently associated with plasma pentosidine levels. These results suggest that liver functional reserve factors (total bilirubin, albumin, and PT-INR) are significant independent factors affecting the plasma pentosidine levels, even when potential cofounders (such as CKD and diabetes) were taken into consideration.

(3) Exclusion criteria mention patients previously treated with GC for 3 months, but do not provide information on or exclude patients with other reasons for osteoporosis, such as postmenopausal women with or without hormone supplementation or patients with other reasons for a substantial vitamin d deficiency. Additionally, did the authors include history of fractures before the primary diagnosis of a liver disease? How did the authors manage patients with CLD and fractures after the age of 40 years that were induced through traumatic injury?

Responses: As pointed out by the reviewer, menopause and vitamin D deficiency (≤20 ng/mL) are known as risk factors for osteoporosis and resultant fractures. Therefore, we newly added the information on menopause and vitamin D deficiency in revised Table 1. However, statistical analyses indicated that the prevalence of vitamin D deficiency did not significantly differ between the fracture and non-fracture groups (revised Table 1) and was not significantly associated with prevalent fractures (revised S2 Table). 

Among the 165 women, 155 (93.9%) were postmenopausal with no hormone supplementation (revised Table 1). Female patients with prevalent fractures had a significantly higher prevalence of menopause than those without prevalent fractures (100% vs. 90.8%, p = 0.019; revised Table 1). However, present study included male patients and most of female patients were postmenopausal. In the future, we are willing to perform the clinical study limited to female patients, including premenopausal women. 

This study included patients with fractures, which occurred after the age of 40 years. However, most of fractures developed in advanced age and after the diagnosis of liver disease. In addition, more than half of vertebral fractures were asymptomatic and diagnosed only by lateral thoracolumbar spine radiographs. Therefore, it is difficult to determine the age of fractures accurately. We newly added the following description in the Method section (lines 121–123): Prevalent vertebral fractures, including asymptomatic fractures diagnosed only by radiography, were semi-quantitatively assessed using lateral thoracolumbar spine radiographs.

We treated fracture patients in collaboration with an orthopedic surgeon. According to the Japanese guidelines for prevention and treatment, patients with vertebral or total hip fractures are diagnosed with osteoporosis, irrespective of BMD values. We usually initiated pharmacological treatment, including bisphosphonate, denosumab, and teriparatide, to prevent occurrence of further fractures in patients with fractures.

(4) Cirrhosis is not sufficiently defined, as scoring systems and diagnostic information are lacking (Child-Pugh, MELD, …). A dichotomous comparison between cirrhotic and non-cirrhotic patients would have been of great interest, as the authors central question for pentosidine measurement involves the differences between sub cohorts of CLD.

Responses: As suggested by the reviewer, we described the definition of liver cirrhosis and Child-Pugh classification in the Method section (lines 106–114). We compared the plasma pentosidine levels among patients with non-LC and LC (new Figure 2). Patients with LC, especially with decompensated LC (Child-Pugh B/C), had significantly higher levels of plasma pentosidine than those with non-LC (new Figure 2A).

(5) A comparison of the different CLD aetiologies would have been interesting. Here, especially the univariate analysis of significant factors associated with prevalent fractures lacks inclusion of CLD

aetiologies. A confounding bias is likely.

Responses: We agree with the reviewer’s comments. We performed univariate regression analysis including variable of CLD aetiology to identify significant factors related to prevalent fractures (revised S2 Table) and CLD aetiology was not significantly associated with prevalent fractures in this study cohort (p = 0.287). 

(6) The characterisation of pentosidine groups has raised important questions, that have not been sufficiently attended. For example, did plasma pentosidine levels correlate with Child-Pugh or MELD in patients with liver cirrhosis? This information would be especially helpful, as laboratory parameters in the different pentosidine groups differed significantly.

Responses: We thank the reviewer for insightful comments. We newly added the

prevalence of decompensated LC (Child-Pugh B/C) in revised Table 3. The H-Pen group had the highest prevalence of LC and decompensated LC among the three groups. Furthermore, plasma pentosidine levels significantly correlated with Child-Pugh scores in patients with LC (new Figure 2B). Accordingly, the H-Pen group showed significantly worse liver functional reserve (total bilirubin, albumin, and PT-INR) compared with the other two groups.

Editor:

Responses: As instructed by the editor, we revised the original manuscript according to PLOS ONE's style requirements.

(2) In your Methods section, please provide additional information about the participant recruitment method and the demographic details of your participants. Please ensure you have provided sufficient details to replicate the analyses such as:

a) a statement as to whether your sample can be considered representative of a larger population, and b) a description of how participants were recruited.

Responses: We agree with your suggestion. Our hospital is located in Fuji City, which has a population of approximately 250,000, near Mt. Fuji, and is the only community-based core hospital (520-bed capacity) in and around Fuji City. Therefore, this study cohort might have heterogeneous clinical characteristics but could represent the actual situation in the community-based and real-world clinical settings. We described these contents in the Methods section (lines 91–95). In addition, we described how to recruit participants in the Methods section (lines 87–91).

(3) PLOS ONE requires experimental methods to be described in enough detail to allow suitably skilled investigators to fully replicate and evaluate your study. To comply with PLOS ONE submission guidelines, in your Methods section, please provide a more detailed description of your methodology in your Biochemical assessment section. Please ensure that you describe the sources and catalog numbers of all ELISA assays, antibodies, etc. in the methods section of your manuscript. For antibodies, please also include the dilutions used in your experiments.

Responses: As instructed by the editor, we described the details of our methodology in the Methods section (lines 131–151). In the present study, plasma pentosidine levels were measured using an ELISA kit (FSK pentosidine ELISA kit; Fushimi Pharmaceutical, Kagawa, Japan). However, the antibody dilution conditions were not listed because they are confidential at Fushimi Pharmaceutical.

---

## [Decision Letter · Decision Letter 1]

16 Mar 2021

PONE-D-21-00527R1

Plasma pentosidine levels are associated with prevalent fractures in patients with chronic liver disease

PLOS ONE

Dear Dr. Saeki,

Thank you for submitting your manuscript to PLOS ONE. After careful consideration, we feel that it has merit but does not fully meet PLOS ONE’s publication criteria as it currently stands. Therefore, we invite you to submit a revised version of the manuscript that addresses the points raised during the review process.

As you can see, both reviewers appreciated the modifications you made and only minor changes are required at this stage.

We look forward to receiving your revised manuscript.

Kind regards,

Pavel Strnad

Academic Editor

PLOS ONE

Journal Requirements:

Reviewers' comments:

Reviewer's Responses to Questions

**Comments to the Author**

1. If the authors have adequately addressed your comments raised in a previous round of review and you feel that this manuscript is now acceptable for publication, you may indicate that here to bypass the “Comments to the Author” section, enter your conflict of interest statement in the “Confidential to Editor” section, and submit your "Accept" recommendation.

Reviewer #1: All comments have been addressed

Reviewer #2: (No Response)

2. Is the manuscript technically sound, and do the data support the conclusions?

Reviewer #1: Yes

Reviewer #2: Yes

3. Has the statistical analysis been performed appropriately and rigorously? 

Reviewer #1: Yes

Reviewer #2: Yes

4. Have the authors made all data underlying the findings in their manuscript fully available?

Reviewer #1: Yes

Reviewer #2: No

5. Is the manuscript presented in an intelligible fashion and written in standard English?

Reviewer #1: Yes

Reviewer #2: Yes

6. Review Comments to the Author

Reviewer #1: In their study the authors determined the relationship between plasma levels of pentosidine and prevalent fractures in 324 patients with chronic liver disease. As mentioned before, the presented data are of relevance, but nevertheless showed a couple of weaknesses in their first manuscript draft.

In the revised version of their work, the authors responded to multiple comments. They included multivariable analyses taking possible confounding factors, such as the presence of CKD and diabetes mellitus into account and newly performed multiple regression analyses to identify significant and independent factors affecting plasma pentosidine levels.

Another major weakness of the study was represented by the great heterogeneity within the cohort regarding different liver disease etiologies. This issue still remains, but the authors now succeeded in improving the characterization of the study cohort and showed that liver disease etiology is not significantly associated with plasma pentosidine levels.

However, the newly supplemented ROC curve analyses unfortunately suggested that plasma pentosidine levels may not be very useful in predicting fractures. The relationship between serum levels of pentosidine and prognosis in patients with liver disease should be investigated in the future.

Overall, the authors have now managed to pay more attention to confounding factors and to better characterise the cohort.

Reviewer #2: Dear Authors,

thank you for providing modifications in the revised protocol. Almost all major and minor concerns have been addressed.

An additional minor revision would raise the overall value:

1. The authors have performed AUROC analysis for the optimal pentosidine cut-off value for predicting prevalent fractures (R2). Please feel encouraged to include and disclose this data in the final submission (at the moment only available for review) in an effort to increase transparency. Additionally, an open discussion on the implications issued by the underlying data is desirable.

7. PLOS authors have the option to publish the peer review history of their article (what does this mean?). If published, this will include your full peer review and any attached files.

Reviewer #1: No

Reviewer #2: No

---

## [Author Response · Author response to Decision Letter 1]

21 Mar 2021

RESPONSES TO THE REVIEWER

We wish to express our deep appreciation to the reviewers for the constructive comments on our manuscript. Our point-by-point responses to the reviewers’ comments are listed below.

Review Comments to the Author:

Reviewer #1: In their study the authors determined the relationship between plasma levels of pentosidine and prevalent fractures in 324 patients with chronic liver disease. As mentioned before, the presented data are of relevance, but nevertheless showed a couple of weaknesses in their first manuscript draft. In the revised version of their work, the authors responded to multiple comments. They included multivariable analyses taking possible confounding factors, such as the presence of CKD and diabetes mellitus into account and newly performed multiple regression analyses to identify significant and independent factors affecting plasma pentosidine levels.

Another major weakness of the study was represented by the great heterogeneity within the cohort regarding different liver disease etiologies. This issue still remains, but the authors now succeeded in improving the characterization of the study cohort and showed that liver disease etiology is not significantly associated with plasma pentosidine levels. However, the newly supplemented ROC curve analyses unfortunately suggested that plasma pentosidine levels may not be very useful in predicting fractures. The relationship between serum levels of pentosidine and prognosis in patients with liver disease should be investigated in the future.

Overall, the authors have now managed to pay more attention to confounding factors and to better characterise the cohort.

Responses: We are deeply grateful for the reviewer’s encouraging comments. We feel the previous reviewers’ comments have helped us markedly improve our manuscript. In the future, we are willing to investigate and report the relationship between plasma pentosidine levels and prognosis in patients with CLD.

Reviewer #2: Dear Authors, thank you for providing modifications in the revised protocol. Almost all major and minor concerns have been addressed.

An additional minor revision would raise the overall value:

1. The authors have performed AUROC analysis for the optimal pentosidine cut-off value for predicting prevalent fractures (R2). Please feel encouraged to include and disclose this data in the final submission (at the moment only available for review) in an effort to increase transparency. Additionally, an open discussion on the implications issued by the underlying data is desirable.

Responses: As suggested by the reviewer, we included the results of AUROC analysis in the final manuscript (S2 Figure). As shown in the first revise, the plasma pentosidine cutoff value for predicting prevalent fractures, AUC, sensitivity, and specificity were 0.0545 μg/mL, 0.60, 0.714, and 0.452, respectively. These results suggest that the plasma pentosidine levels are not very useful for predicting prevalent fractures in our cross-sectional study. In the furture, a large-scale multicenter, prospective studies are needed to conclude the usefulness of plasma pentosidine in predicting the occurrence of further fragility fractures.

---

## [Editor Report · Decision Letter 2]

24 Mar 2021

Plasma pentosidine levels are associated with prevalent fractures in patients with chronic liver disease

PONE-D-21-00527R2

Dear Dr. Saeki,

We’re pleased to inform you that your manuscript has been judged scientifically suitable for publication and will be formally accepted for publication once it meets all outstanding technical requirements.

Kind regards,

Pavel Strnad

Academic Editor

PLOS ONE

Additional Editor Comments (optional):

Thanks for your nice contribution!
---

## [Editor Report · Acceptance letter]

26 Mar 2021

PONE-D-21-00527R2 

Plasma pentosidine levels are associated with prevalent fractures in patients with chronic liver disease 

Dear Dr. Saeki:

I'm pleased to inform you that your manuscript has been deemed suitable for publication in PLOS ONE. Congratulations! Your manuscript is now with our production department. 

Kind regards, 

on behalf of

Dr. Pavel Strnad 

Academic Editor

PLOS ONE